# Utility of Geriatric Nutritional Risk Index in Patients with Chronic Kidney Disease: A Mini-Review

**DOI:** 10.3390/nu13113688

**Published:** 2021-10-20

**Authors:** Naoki Nakagawa, Keisuke Maruyama, Naoyuki Hasebe

**Affiliations:** Division of Cardiology, Nephrology, Pulmonology and Neurology, Department of Internal Medicine, Asahikawa Medical University, Asahikawa 078-8510, Japan; maru824@asahikawa-med.ac.jp (K.M.); haselove@asahikawa-med.ac.jp (N.H.)

**Keywords:** frailty, geriatric nutritional risk index, kidney disease, malnutrition, sarcopenia

## Abstract

Chronic kidney disease (CKD) is one of the most significant risk factors for cardiovasculardisese. Malnutrition has been recognized as a significant risk factor for cardiovascular disease in patients with CKD, including those on chronic dialysis. Current studies showed higher all-cause and cardiovascular mortality rates in patients with CKD and malnutrition. Geriatric nutritional risk index (GNRI), a simple and validated nutritional screening measure for both elderly people and patients on dialysis, is based only on three objective parameters: body weight, height, and serum albumin level. Recently, we demonstrated that the cutoff GNRI for predicting all-cause and cardiovascular mortality was 96 in patients on hemodialysis. Moreover, together with left ventricular hypertrophy and low estimated glomerular filtration rate, the utility of GNRI as a significant determinant of cardiovascular events was demonstrated in non-dialysis-dependent patients with CKD. In the present review, we summarize available evidence regarding the relationship of GNRI with all-cause and cardiovascular mortality in patients with CKD including those on dialysis.

## 1. Introduction

Malnutrition, a common problem in patients with end-stage renal diseases (ESRD) [1,2], is related with higher risk of mortality [2,3] and well-known as the malnutrition inflammation-atherosclerosis [4]. Regular nutritional assessment, such as the determination of body mass index (BMI) and serum albumin, is recommended in all patients on dialysis to decrease mortality [3]. Moreover, protein-energy wasting, which is a form of malnutrition characterized by reduced body protein and energy as a result of catabolic inflammatory responses, happens often in elderly people and patients with ESRD [5]. Among the validated measures until now, malnutrition inflammation score (MIS) is the most extensively validated screening measure for patients on hemodialysis at nutritional risk [6]. However, the malnutrition inflammation score requires subjective assessment by highly-trained examiners to verify consistent results. Geriatric nutritional risk index (GNRI), which is based only on body weight, height, and serum albumin level, has been recently demonstrated as a very simple and objective tool to assess nutritional status in various pathological conditions [7,8]. Until now, several studies have examined the credibility of GNRI in assessing malnutrition and predicting all-cause mortality and cardiovascular (CV) events in patients on chronic hemodialysis [2,8,9,10,11,12,13,14,15,16], peritoneal dialysis (PD) [17,18,19], non-dialysis-dependent (NDD) chronic kidney disease (CKD) [20,21], stroke [22,23,24,25], and heart failure [26,27,28].

Nutritional management is at the center of high-quality treatment in patients with CKD. In this review, we summarize available evidence about the relationship of GNRI with mortality and CV events in patients with CKD including those on dialysis.

## 2. Utility of GNRI as a Simple Nutritional Assessment

The measures for validating nutritional assessment of elderly patients on hemodialysis is consistent with the clinical practice guidelines on the management of elderly patients with advanced CKD, such as stage 3b or higher [29]. These guidelines emphasize the importance of conducting studies whether the nutritional assessment measures suggested by the expert panel of the International Society in Renal Nutrition and Metabolism for adult patients on hemodialysis are related with higher rates of mortality in elderly patients on hemodialysis [30]. Among nutritional parameters, the subjective global assessment (SGA), based on medical history and clinical findings, is an authorized clinical measure for screening nutrition at risk [31]. This measure is currently used in a wide variety of health care situation and also used as a benchmark for developing other nutritional screening measures for patients with CKD [31]. Kalantar-Zadeh et al. developed the MIS, which involves seven parameters from the SGA and three additional non-SGA parameters—BMI, serum albumin, and total iron-binding capacity [32]. The MIS has been validated as a better nutritional assessment measure than the SGA for patients on both chronic hemodialysis and PD [32,33]. However, because the MIS and SGA need subjective assessment and judgment by the highly-trained examiner to verify consistent results among different examiners and at different times. Yamada et al. reported that GNRI has the highest accuracy with the MIS compared to several other nutritional screening measures [8], whereas Szeto et al. found that the GNRI was significantly correlated with other nutritional measures in patients on PD [17]. Therefore, the GNRI has been attracting attention as a good predictor of prognosis in patients with CKD.

The GNRI, originally developed to predict malnutrition related complications and mortality in hospitalized elderly patients, is calculated as follows: GNRI = 14.89 × serum albumin (g/dL) + [41.7 × body weight/ideal body weight] [7]. Body weight/ideal body weight is set to 1 if the body weight exceeds the ideal body weight. Instead of using the Lorentz formula in the original GNRI equation [7], the ideal body weight in recent studies was defined as the value calculated from the height and a BMI of 22 kg/m^2^, as described previously [8]. The cutoff GNRI values according to weight and albumin level were initially defined as follows: <82, major nutrition-related risk; between 82 and <92, moderate nutrition-related risk; between 92 and ≤98, low nutrition-related risk; >98, no risk [7].

Some studies have examined the GNRI in patients on hemodialysis. One study reported that 31.6% of patients on hemodialysis were malnourished according to the GNRI [34]. Current available evidence shows that the GNRI has widely used as a nutrition assessment measure with good performance in patients on dialysis. Beyond its role as a screening measure for malnutrition, Kobayashi et al. also demonstrated that the GNRI was a significant predictor for mortality in patients on hemodialysis [9]. Their findings were subsequently confirmed by other studies, including those examining Asian patients on chronic hemodialysis [9,10,12,13,14,15] as well as other studies from Italy, Netherlands, and France [11,35,36]. In a recent prospective cohort study comprising 3436 Japanese patients on hemodialysis, Matsukuma et al. reported that a lower GNRI was an independent risk factor for infection-related and all-cause mortality in patients on hemodialysis [15]. Another study with a large Japanese cohort also demonstrated that a low/medium GNRI was related with all-cause mortality [16]. The relationship between the GNRI and mortality has been also validated in patients on PD and in those with NDD-CKD [18,19,21]. A recent meta-analysis further confirmed the findings of earlier studies reporting that a high GNRI could reduce the risk of all-cause mortality and that a lower GNRI was related with higher mortality in patients on dialysis [2].

BMI and serum albumin are the two parameters used to calculate the GNRI. A study previously demonstrated that both low body weight and hypoalbuminemia could reflect malnutrition and that lower BMI could also be an important index of protein-energy wasting [37]. BMI is related with mortality in patients on hemodialysis, as previously reported [15]. Additionally, some studies have demonstrated that significant relationships of body weight and height, alone or in combination, are related with all-cause mortality [38,39]. Both BMI and albumin is strong predictors of mortality. A Japanese study demonstrated that the GNRI was superior to BMI and albumin alone in predicting CV disease (CVD) mortality [12]. Thus, the GNRI, which combines BMI and albumin, might have critical utility in predicting the nutrition condition and mortality risk of patients. Chronic inflammation, which occurs frequently in patients on dialysis, may be closely link to increased mortality via malnutrition [13]. Additionally, patients on hemodialysis with a lower GNRI had reduced hemoglobin and albumin levels as well as reduced body weight [15,16]. One study also indicated that patients with a lower GNRI were correlated with poor response to treatment with erythropoietin [40]. Additionally, a low GNRI was associated with low lean mass index, especially in female patients [41]. Inflammation, anemia, and lean mass index are all contributors of malnutrition. Thus, the GNRI is used for the assessment of nutritional condition and the prediction of nutrition-related complications. A low GNRI indicates malnutrition whereas a good nutritional condition is a strong indicator of survival; therefore, we speculate that in patients on hemodialysis the low GNRI may predicate mortality, which is primarily associated with malnutrition. Recent epidemiologic studies have also utilized the GNRI to predict CVD outcomes [26] and have shown that GNRI is independently related with CV events in patients with chronic heart failure [27,42]. Several recent studies have investigated whether GNRI could accurately predict all-cause and CV mortality in patients on hemodialysis [9,10,11,12,13,14,15,16] (Table 1).

## 3. Relationship between GNRI, Inflammation, and Mortality in Patients on Hemodialysis

Inflammation is a potential contributor to CVD, whereas C-reactive protein (CRP) is a significant marker of inflammation associated with CV mortality in patients on hemodialysis [43,44]. Further, the combination of GNRI and CRP increases their predictive value for the risk of CVD-related and all-cause mortality in patients on hemodialysis [14]. Whereas prior studies demonstrated the close relationship between aortic calcification and CV mortality, one study demonstrated that the GNRI was an independent risk factor for the progression of aortic calcification in patients on hemodialysis [45].

We previously examined the relationship between GNRI and CVD in a cohort of 133 patients on hemodialysis with a mean age of 59.8 ± 10.2 years [13]. Of the cohort, 57.9% were male, and the rates of diabetes and metabolic syndrome were 26.3% and 38.3%, respectively. During the 6-year observation period, 41 (30.9%) patients deceased. The cause of death was as follows; heart failure (*n* = 13), acute myocardial infarction (*n* = 11), stroke (*n* = 2), infection (*n* = 10), and cancer (*n* = 5). The duration of hemodialysis, sex, and adequacy of dialysis as indicated by Kt/V urea were not significantly different between the two groups. Compared with the surviving patients, the deceased patients were significantly older (64.5 ± 8.3 vs. 57.7 ± 10.2 years, *p* < 0.001) and had a higher frequency of diabetes (44% vs. 18%, *p* = 0.002), higher levels of tumor necrosis factor (TNF)-α (5.0 ± 1.8 vs. 4.0 ± 1.6 pg/mL, *p* = 0.001) and high-sensitivity CRP (hs-CRP) (0.47 ± 0.65 vs. 0.18 ± 0.29 mg/dL, *p* = 0.006), and higher aortic calcification scores (42.9 ± 23.0 vs. 29.9 ± 23.8, *p* = 0.004). Additionally, the deceased patients had lower serum albumin levels (3.7 ± 0.2 vs. 3.9 ± 0.3 g/dL, *p* < 0.001) and GNRI (94.5 ± 4.4 vs. 97.3 ± 5.2, *p* = 0.001), suggesting that the deceased patients exhibited more severe malnourishment and worse inflammation than the surviving patients. Furthermore, the receiving-operating characteristic (ROC) curve analysis of 6-year all-cause mortality showed significantly increased all-cause and CV mortality in patients with a GNRI of ≤96 compared to those with a GNRI of >96, suggesting that a GNRI of ≤96 might be a predictor of mortality in patients on hemodialysis, although this cutoff was higher than that reported in previous studies [8,9,10].

Furthermore, we investigated the impact of TNF-α and hs-CRP on mortality separated by the presence of malnutrition, which was defined as a GNRI of ≤96 based on the above results. Based upon the ROC curve analysis for the prediction of all-cause mortality at the end of the observation period, the optimal TNF-α and hs-CRP cutoff values were 3.5 pg/mL and 0.04 mg/dL, respectively. Higher TNF-α and hs-CRP levels were related with higher all-cause (*p* = 0.002 and *p* = 0.030, respectively) and CV mortality (*p* = 0.003 and *p* = 0.042, respectively) in malnourished patients, suggesting that these inflammation markers were useful as predictors of all-cause and CV mortality particularly in malnourished patients as well as shown in the Framingham Offspring cohort [46].

## 4. Synergistic Impact of GNRI and Left Ventricular Hypertrophy on Patients with NDD-CKD

A study in patients with NDD-CKD found that a lower GNRI was independently associated with progression to ESRD and that this parameter might be useful for predicting the poor renal outcomes in patients with stage 3–5 CKD [47]. Another study showed that the GNRI was a good predictor of muscle function in renal transplant recipients [48]. Furthermore, the GNRI is currently considered as a prognostic factor in patients with cancer [49]. Overall, multiple lines of evidence demonstrate that the GNRI can be a favorable predictor for clinical outcomes in a variety of distinct diseases and may be wide utility in clinical setting.

Whereas malnutrition shows poor prognoses in various clinical settings, the synergistic impact of nutritional parameters, left ventricular hypertrophy (LVH), and renal dysfunction on CV events is unknown. Therefore, we investigated the associations among the GNRI, LVH, estimated glomerular filtration rate (eGFR), and CV events in patients including those with NDD-CKD [20]. Among a cohort of 338 patients who underwent echocardiographic evaluation, we examined 161 patients who were followed up for more than seven years. The mean age, eGFR, left ventricular mass index (LVMI), and GNRI were 63.5 ± 9.2 years, 72.9 ± 18.7 mL/min/1.73 m^2^, 114 ± 33 g/m^2^, and 100.4 ± 6.0, respectively. Within the cohort, 25% (*n* = 40) of the patients had an eGFR of <60 mL/min/1.73 m^2^. In addition, 15 (9%) patients had proteinuria, 46 (29%) patients had CKD, and 59 (37%) patients had LVH, which was defined as an LVMI of ≥125 g/m^2^ in males and ≥ 110 g/m^2^ in females. During the observation period, 15 (9.3%) patients experienced CV events as follows; acute coronary syndrome (*n* = 7), heart failure (*n* = 4), stroke (*n* = 2), aortic dissection (*n* = 1), and aortic rupture (*n* = 1). In univariate regression analysis, we found that eGFR exhibited a significant negative correlation with age, proteinuria, left atrial diameter (LAD), and LVMI and a significant positive correlation with serum albumin level, GNRI, and early diastolic velocity of mitral annular motion (e’). LVMI had a significant negative association with eGFR and e’ and a positive association with the prevalence of proteinuria, and LAD. Meanwhile, the GNRI correlated significantly and positively with BMI, hemoglobin and serum albumin levels, and eGFR, but did not correlate with LAD, LVMI, and left ventricular end-diastolic diameter, suggesting the GNRI was irrespective of volume overload. Furthermore, we investigated independent parameters for CV events using the Cox proportional hazards model. We found that age (hazard ratio [HR] 1.121, 95% confidence interval [CI] 1.013–1.241), eGFR (HR 0.961, 95% CI 0.928–0.995), LVMI (HR 1.017, 95% CI 1.002–1.033) and GNRI (HR 0.886, 95% CI 0.807–0.973) were significantly related with CV events after adjusting for sex, hemoglobin level, prevalence of hypertension, and prevalence of diabetes, suggesting that malnutrition was an independent predictor of CV events even in patients with NDD-CKD.

Next, we investigated the impact of malnutrition, LVH, and low eGFR on CV events using the Kaplan–Meier analysis by stratifying the patients according to the presence of malnutrition, which was defined as a GNRI of ≤96 as previously reported [13]. We found that lower eGFR, lower GNRI, the presence of LVH and proteinuria, and higher LAD were significantly related with higher incidence of CV events. In contrast, the prevalence of hypertension was not significantly related with CV events. Furthermore, we examined the synergistic impact of low GNRI, LVH, and low eGFR on CV events and found that the combination of the presence of LVH and lower eGFR was significantly related with higher rates of CV events (*p* = 0.003). Additionally, the combination of lower GNRI and lower eGFR was also significantly related with higher incidence of CV events (*p* < 0.001). Furthermore, the combination of lower GNRI and the presence of LVH was significantly related with higher incidence of CV events not only in all patients (*p* < 0.001) but also in those with CKD (n = 46, *p* = 0.014). Overall, these results suggest that malnutrition, LVH, and renal dysfunction have a synergistic impact on the frequency of CV events also in patients with NDD-CKD.

It is well known that age, hypertension, LVM, and LAD are related to the frequency of atrial fibrillation [50]. It is possible that these factors might mediate the relationship between CKD and atrial fibrillation incidence observed in several previous studies [51,52], because eGFR gradually decreases with age whereas pressure and volume load directly increase arterial stiffness, LAD, and LVM [53,54]. A recent study has linked subclinical nephrosclerosis to LVH independently of classical risk factors for atherosclerosis [55]. Furthermore, there is a strong correlation among malnutrition, inflammation, and atherosclerosis, three significant independent clinical entities that coexist in patients with ESRD and are well known as malnutrition inflammation-atherosclerosis syndrome [4], suggesting that chronic inflammation might have affected CV events in our previous study [13] and others [46,56].

In summary, we clearly demonstrated that low GNRI, LVH, and low eGFR were independent variables of CV events and that they were associated with an increase in these events in the long-term. Furthermore, we clearly showed that, although a cutoff GNRI of ≤ 96 might be useful in predicting CV events even in patients with NDD-CKD, the best GNRI cutoff might be different across distinct ethnic populations. Despite the variability in the precise cutoff GNRI for the prediction of all-cause mortality, a lower GNRI is clearly related with greater risk. Future studies are warranted to determine the best GNRI cutoff value o for morality prediction in patients with CKD.

## 5. Conclusions

As illustrated in the present review, we provide evidence that malnutrition increases the incidence of all-cause and CV mortality in patients with CKD, with the aim to illustrate the importance of recognizing nutritional condition and to highlight the GNRI as a simple and powerful measure to evaluate malnutrition in the clinical setting. Appropriate management of malnutrition is critical to improve survival in patients with CKD.

## Figures and Tables

**Table 1 nutrients-13-03688-t001:** Summary of cohort studies investigating the relationship of GNRI with outcomes in patients with CKD.

Patients	Authors, Year	Country	Age (Year)	N, Male (%)	Follow-Up Time	Cutoff Value of GNRI	Outcomes	Ref.
Hemodialysis	Kobayashi et al., 2010	Japan	60 ± 12	490, 59.8%	60 months	90	All-cause mortality	[8]
	Park et al., 2012	Korea	56.2 ± 12.7	120, 42.5%	120 months	90	All-cause mortality	[9]
	Panichi et al., 2014	Italy	65.7 ± 14.1	753, 60.7%	84 months	92	All-cause mortality	[10]
	Takahashi et al., 2014	Japan	64 ± 13	1568, 66.9%	63 months	92	All-cause and CV mortality	[11]
	Nakagawa et al., 2015	Japan	59.8 ± 10.2	133, 57.9%	72 months	96	All-cause and CV mortality	[12]
	Ishii et al., 2017	Japan	64 ± 12	973, 62.6%	96 months	91.2	All-cause and CV mortality	[13]
	Matsukuma et al., 2019	Japan	63.7 ± 12.8	3436, 59.1%	48 months	95.8	All-cause and infection-related mortality	[14]
	Yamada et al., 2020	Japan	66 (58–74) *	3536, 65%	26 months	89.3	All-cause mortality	[15]
Peritoneal dialysis	Kang et al., 2013	Korea	52.5 ± 15.1	486, 53.1%	36 months	96.4	All-cause mortality	[17]
	Ren et al., 2020	China	50.2 ± 14.4	1804, 55.4%	33.7 months	94.55	All-cause mortality	[18]
NDD-CKD	Maruyama et al., 2016	Japan	63.5 ± 9.2	161, 51%	96 months	96	CV events	[19]
	Kiuchi et al., 2016	Japan	67 (37–81) ^#^	126, 51.6%	64 months	92	All-cause mortality and CV events	[20]

CKD, chronic kidney disease; CV, cardiovascular; GNRI, Geriatric nutritional risk index; NDD, non-dialysis-dependent, * median (95% CI), ^#^ median (10th–90th percentile).

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
