# Peer review of "Utility of Geriatric Nutritional Risk Index in Patients with Chronic Kidney Disease: A Mini-Review"

_nutrients, 2021, doi:10.3390/nu13113688_

Round 1

Reviewer 1 Report

  1. In line 47 to 49, the authors cited reference 6 for the equation of GNRI: 14.89 × serum 48 albumin (g/dL) + 41.7 × BMI/22. However, in the original article, the Lorentz formula was applied, instead of “BMI/22”.
  2. In line 78 to 80, the authors cited reference 28 to illustrate the association between anthropometric measurements and all-cause morbidity and mortality. However, the cited article is a cross-sectional study, investigating the correlation between nutritional markers and appetite self-assessments in hemodialysis patients.
  3. It would be more informative if important epidemiologic data, such as sex, mean age, ethnicity, country, dialysis vintage and follow-up period, could be added in the Table 1.
  4. In line 112, the authors cited reference 38 showing GNRI as an independent risk factor for progression of aortic calcification in patients with NDD-CKD. However, the original article is a cross-sectional study. It may be inappropriate to be used as the reference here.
  5. In line 132, it is unclear why the authors state that “…was based on patients at moderate nutrition-related risk.” in the context of the paragraph.
  6. In section 3, the authors put emphasis on the correlation between GNRI, malnutrition inflammation-atherosclerosis syndrome (MIAS), and cardio-renal-anemia syndrome (CRAS). However, they largely described the association between malnutrition and MIAS. None of evidence regarding CRAS was shown.
  7. In section 5, the authors discussed “Management of patients with CKD using the GNRI”. However, the content is about the cutoffs of GNRI as the predictor for clinical outcomes, which seemed duplicated from the Table 1.

Reviewer 2 Report

Dear Authors,

you should more extensively discuss 

the relationship between GNRI and other

Malnutrition scores such as the SGA, the MIS

Author Response

We thank the reviewer for these valuable comments and suggestions. We have provided additional explanations and expanded the discussion in accordance with the editor’s and reviewer’s comments.

Point 1:  You should more extensively discuss the relationship between GNRI and other malnutrition scores such as the SGA, the MIS.

Response 1: We appreciate the reviewer’s suggestion. We have modified our revised manuscript as this reviewer’s suggestion. Please see line 47 to 67, page 2 in the revised manuscript.

Round 2

Reviewer 1 Report

The authors provided a revised version for review of the association between GNRI and clinical outcomes in patients with CKD. They have made some emendations according to the suggestions. However, the main problems remained. The discussion was primarily derived from their own studies (reference 12, 19). It would be more appropriate to cite others’ articles, in addition to their own works, in this review article. In addition, the evidence supporting the other subjects seemed very weak, such as the association between GNRI and the cardio-renal-anemia syndrome. The section 5, “Management of patients with CKD using the GNRI”, was still duplicated from the previous Table 1. Overall, the work is not very well organized and comprehensively described.
